# Genomic basis for drought resistance in European beech forests threatened by climate change

**Markus Pfenninger[1,2,3]\*, Friederike Reuss[1], Angelika Kiebler[1], Philipp Schönnenbeck[1,4], Cosima Caliendo[1,4], Susanne Gerber[4], Berardino Cocchiararo[3,5], Sabrina Reuter[6], Nico Blüthgen[6], Karsten Mody[6,7], Bagdevi Mishra[8], Miklós Bálint[3,9,10], Marco Thines[3,8,11], Barbara Feldmeyer[1]**

[1]Molecular Ecology, Senckenberg Biodiversity and Climate Research Centre, Frankfurt am Main, Germany; [2]Institute for Organismic and Molecular Evolution, Johannes Gutenberg University, Mainz, Germany; [3]LOEWE Centre for Translational Biodiversity Genomics, Frankfurt am Main, Germany; [4]Institute of Human Genetics, University Medical Center, Johannes Gutenberg University, Mainz, Germany; [5]Conservation Genetics Section, Senckenberg Research Institute and Natural History Museum Frankfurt, Gelnhausen, Germany; [6]Ecological Networks lab, Department of Biology, Technische Universität Darmstadt, Darmstadt, Germany; [7]Department of Applied Ecology, Hochschule Geisenheim University, Geisenheim, Germany; [8]Biological Archives, Senckenberg Biodiversity and Climate Research Centre, Frankfurt am Main, Germany; [9]Functional Environmental Genomics, Senckenberg Biodiversity and Climate Research Centre, Frankfurt am Main, Germany; [10]Agricultural Sciences, Nutritional Sciences, and Environmental Management, Universität Giessen, Giessen, Germany; [11]Institute for Ecology, Evolution and Diversity, Johann Wolfgang Goethe-University, Frankfurt am Main, Germany

**\*For correspondence:**
Markus.Pfenninger@senckenberg.de

**Competing interests:** The authors declare that no competing interests exist.

**Abstract** In the course of global climate change, Central Europe is experiencing more frequent and prolonged periods of drought. The drought years 2018 and 2019 affected European beeches (*Fagus sylvatica* L.) differently: even in the same stand, drought-damaged trees neighboured healthy trees, suggesting that the genotype rather than the environment was responsible for this conspicuous pattern. We used this natural experiment to study the genomic basis of drought resistance with Pool-GWAS. Contrasting the extreme phenotypes identified 106 significantly associated single-nucleotide polymorphisms (SNPs) throughout the genome. Most annotated genes with associated SNPs (>70%) were previously implicated in the drought reaction of plants. Non-synonymous substitutions led either to a functional amino acid exchange or premature termination. A non-parametric machine learning approach on 98 validation samples yielded 20 informative loci which allowed an 88% prediction probability of the drought phenotype. Drought resistance in European beech is a moderately polygenic trait that should respond well to natural selection, selective management, and breeding.

## Introduction

Climate change comes in many different facets, amongst which are prolonged drought periods (*Christensen et al., 2007*). The Central European droughts in the years 2018 and 2019 caused severe water stress in many forest tree species, leading to the die-off of many trees (*Schuldt et al.,*

**eLife digest** Climate change is having a serious impact on many ecosystems. In the summer of 2018 and 2019, around two thirds of European beech trees were damaged or killed by extreme drought. It is critical to keep these beech woods healthy, as they are central to the survival of over 6,000 other species of animals and plants.

The level of damage caused by the drought varied between forests. However, not all the trees in each forest responded in the same way, with severely damaged trees often sitting next to fully healthy ones. This suggests that the genetic make-up of each tree determines how well it can adapt to drought rather than its local environment.

To investigate this further, Pfenninger et al. studied the genome of over 400 European beech trees from the Hesse region in Germany. The samples came from pairs of neighbouring trees that had responded differently to the droughts. The analysis found more than 80 parts of the genome that differed between healthy and damaged trees.

Pfenninger et al. then used this information to create a genetic test which can quickly and inexpensively predict how well an individual beech tree might survive in a drought. Applying this test to another 92 trees revealed that it can reliably detect which ones were healthy and which ones were damaged.

Beech forests are typically managed by private owners, agencies or breeders that could use this genetic test to select and reproduce trees that are better adapted to drought. The goal now is to develop the test so that it can be used more widely to manage European beech trees and potentially other species.

*2020*). Among the suffering tree species was European beech, *Fagus sylvatica* L. As one of the most common deciduous tree species in Central Europe, *F. sylvatica* is of great ecological importance: beech forests are a habitat for more than 6000 different animal and plant species (*Brunet et al., 2010*; *Dorow et al., 2010*). The forestry use of beech in 2017 generated a turnover of more than €1 billion in Germany alone (*Thünen_Institute, 2020*), without taking the economic and societal value of the ecosystem services of woods into account (*Elsasser et al., 2016*). However, the drought years 2018 and 2019 severely impacted the beech trees in Germany (*Paar and Dammann, 2019*). Official reports on drought damage in beech recorded 62% of trees with rolled leaves and 20–30% of small leaves, mainly in the crown, resulting in 7% of badly damaged or dead trees. As shown before (*Bressem, 2008*), mainly medium- to old-aged trees were affected by drought stress (>60 years).

Under favourable conditions, beech is a competitive and shade-tolerant tree species, dominating mixed stands (*Pretzsch et al., 2013*). High genetic diversity within populations supports adaptation to local conditions (*Kreyling et al., 2012*). Significant differences between local populations in tolerance to various stress factors such as early frost (*Czajkowski and Bolte, 2006*), drought (*Cocozza et al., 2016*; *Harter et al., 2015*), or air pollution (*Müller-Stark, 1985*) are known. The distribution of *F. sylvatica* is mainly limited by water availability as the tree does not tolerate particularly wet or dry conditions (*Sutmöller et al., 2008*). Therefore, it is quite conceivable that the species could suffer even more under the predicted future climatic conditions than today (*Sutmöller et al., 2008*).

Despite the widespread, severe drought damage, a pattern observed in all beech forests was very noticeable (personal observations). Using crown deterioration as a significant indicator for drought damage (*Choat et al., 2018*), not all trees in a beech stand were equally damaged or healthy. The damage occurred rather in a mosaic-like pattern instead. Even though the extent of drought damage varied among sites, apparently completely healthy trees immediately neighboured severely damaged ones and vice versa. This observation gave rise to the hypothesis that not the local environmental conditions might be decisive for the observed drought damage, but rather the genetic make-up of the individual trees.

We decided to draw on this natural 'experimental set-up' to infer the genomic basis underlying the drought susceptibility in *F. sylvatica*. We identified more than 200 neighbouring pairs of trees with extreme phenotypes and used a Pool-GWAS approach (*Bastide et al., 2013*) to infer associated single-nucleotide polymorphism (SNP) loci by contrasting allele frequencies with replicated pools of

drought-susceptible and -resistant individuals. In addition, we individually resequenced a subset of 51 pairs of susceptible and resistant trees. If the observed pattern indeed has a genetic basis, identifying the associated loci would enable the genomic prediction of drought resistance (*Stocks et al., 2019*). Constructing an SNP assay from the most highly phenotype-associated SNPs, we validated 70 identified loci by predicting the drought phenotype of an additional set of beech trees from their genotype at these loci using linear discriminant analysis (LDA) and a new Machine Learning approach (*Horenko, 2020*). These accurate genomic prediction tools, for example, the choice of drought-resistant seed-producing trees and selective logging, could help accelerate and monitor natural selection and thus harness beech forests against climate change (*Waldvogel et al., 2020*).

## Results

### Sampling, climate development, and phenotyping

Damaged and healthy beech tree pairs were sampled from woods in the lowland Rhein-Main plain, the adjacent low mountain ranges of Odenwald and Taunus, and mountain ranges from Central and Northern Hessen (*Figure 1A*). When summarising the climatic conditions from 1950 to 2019 for the sampling sites in a principal component analysis (PCA), the sites were divided into two groups by axis 1, a temperature gradient. The Taunus mountain sites grouped with those from the northern part of Hessen, while the Rhein-Main plain clustered with the Odenwald sites (*Figure 1B*). This grouping was also used to construct the GWAS pools (see below). Comparing the climate from the 1950s, when most of the trees sampled were already in place, with the decade from 2010 to 2019, showed that all local conditions changed substantially and similarly in the direction and extent of warmer and drier conditions (*Figure 1B*). The steepest temperature increase occurred in the 1980s, while precipitation patterns mainly changed in the last decade (*Figure 1—figure supplement 1*). A wide range of parameters, potentially relevant as selection pressures, changed drastically during this period: the mean January daily minimum temperature at the sampling sites increased by 1.49°C from 2.64°C (s.d. 1.68°C) in the 1950s to 1.15°C (s.d. 2.50°C) during the last decade. The mean August daily maximum temperatures increased even more by 2.37°C from 22.06°C (s.d. 1.95°C) to 24.43°C (s.d. 2.35°C). Simultaneously, mean annual precipitation decreased by 40.5 mm or 5.5% from 741.2 mm (s.d. 85.8 mm) to 700.7 mm (s.d. 70.9 mm). Most of the precipitation loss (84%) occurred during the main growth period between April and September, with a decrease of 33.9 mm from 410.4 mm (s.d. 36.1 mm) to 376.5 mm (s.d. 25.6 mm).

Mean monthly evaporation potential, available from 1991 onwards, showed that, compared to the beginning of the 1990s, the main growth period of beech from April to September became increasingly drier, with up to 30 mm more evaporation per month. The drought dynamics suggested that the years 2018 and 2019 were not outliers, but rather part of a long-term, accelerating trend (*Figure 1C*), following the overall global pattern (*Büntgen et al., 2021*; *Trenberth et al., 2014*).

There was a strong negative correlation (r = 0.695) between the drought strength during the main growth period (Apr– Sept) and a proxy for (green) leaf cover (leaf area index [LAI]) for the sampled plots in the years 2015–2019 (*Figure 1—figure supplement 2*). This observation suggested that leaf loss and dried leaves are good indicators for drought stress.

The mean distance between paired trees was 5.1 m (s.d. 3.4 m, *Figure 2—figure supplement 1*). Phenotypic measurements generally confirmed the study design and selection of trees: healthy and damaged trees within each tree pair did not differ significantly in *trunk circumference*, *tree height*, *canopy closure*, and *competition index* (*Figure 2A–D*, *Supplementary file 1B*). Hence, these parameters were not considered in further analyses. As expected, and confirming the assignment of damage status, the quantity of *dried leaves* and *leaf loss* differed substantially between damaged and healthy ones (*Figure 2E, F*, *Supplementary file 1B*). A sample of photographs contrasting damaged and healthy paired trees can be found in *Figure 2—figure supplement 2*.

### Linkage disequilibrium, population structure, and genome-wide association study

For a subsample of 300 out of the 402 sampled beech trees, we generated four DNA pools from two climatically distinct regions (North and South Hessen, *Figure 1B*), contrasting trees that were either healthy or highly drought damaged, respectively (*Supplementary file 1A*). The 'South' pools

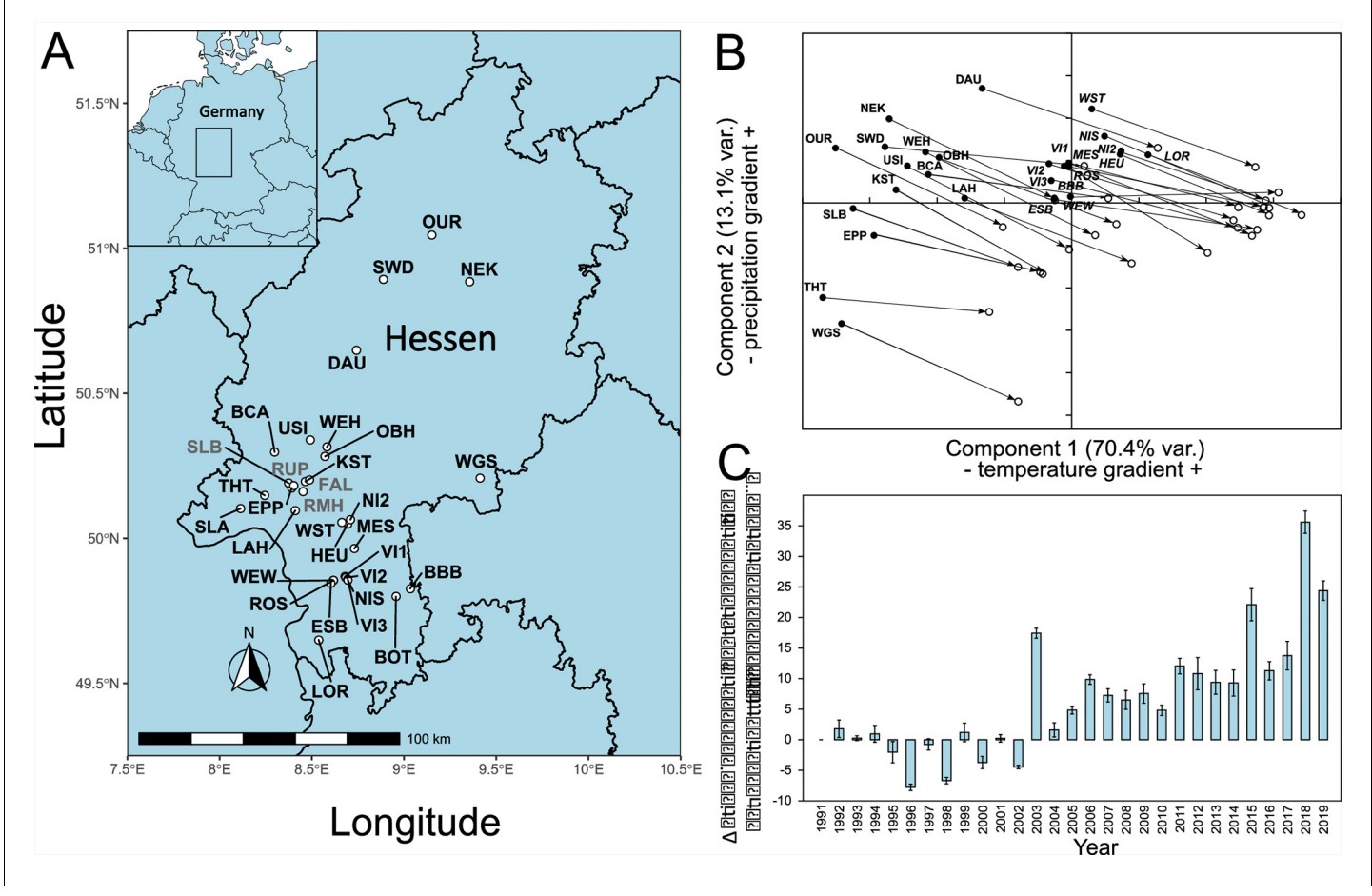

**Figure 1.** Location of sampling sites and climate history. (**A**) Locations of sampling sites in Hessen, Germany. For abbreviations, see *Supplementary file 1A*. The sites where confirmation individuals were sampled are designated in grey. (**B**) Principal component analysis of monthly climate data 1950–2019. (**C**) Development of main growth period drought indicator from 1991 to 2019. Shown is the difference mean monthly evaporation potential in mm from April to September relative to the 1991 level. Climate and drought data obtained from https://opendata.dwd.de/ climate_environment/CDC/grids_germany/monthly/.

The online version of this article includes the following figure supplement(s) for figure 1:

**Figure supplement 1.** Climate change dynamics.

**Figure supplement 2.** Plot of 'relative leaf area index' change relative to 2014 values against 'cumulated evatransporation potential' change during the growth season relative to 2014 values for all 1 1 km plots encompassing the 27 sampling sites.

consisted of 100 individuals each, whereas the 'North' pools contained 50 individuals each. We created ~50 GB 150 bp-paired end reads with insert size 250–300 bp on an Illumina HiSeq 4000 system per pool. More than 96% of the reads mapped against the repeat-masked chromosome-level beech reference genome (accession no. PRJNA450822). After filtering the alignment for quality and a coverage between 15 and 70 , and removing indels, allele frequencies for 9.6 million SNPs were scored. All 100 individuals from the North population were additionally individually resequenced to ~20 coverage each (for more details, see Materials and methods). This data was used to (i) determine individual variability in allele frequencies and (ii) validate the information content of the candidate SNP set.

Using all individually resequenced individuals, we inferred the extent of genome-wide linkage disequilibrium (LD). The plot of LD $r^2$ against the distance from the focal SNP showed that LD fell to $r^2 \sim 0.3$ within <120 bp, which means that genome positions such a distance apart are on average effectively unlinked (*Figure 3—figure supplement 1A*). The PCA on SNP variation of the individually resequenced trees from the North population explained 12.3% of accumulated variation on the first two axes (*Figure 3—figure supplement 1B*). Trees from the same sampling site (within the North

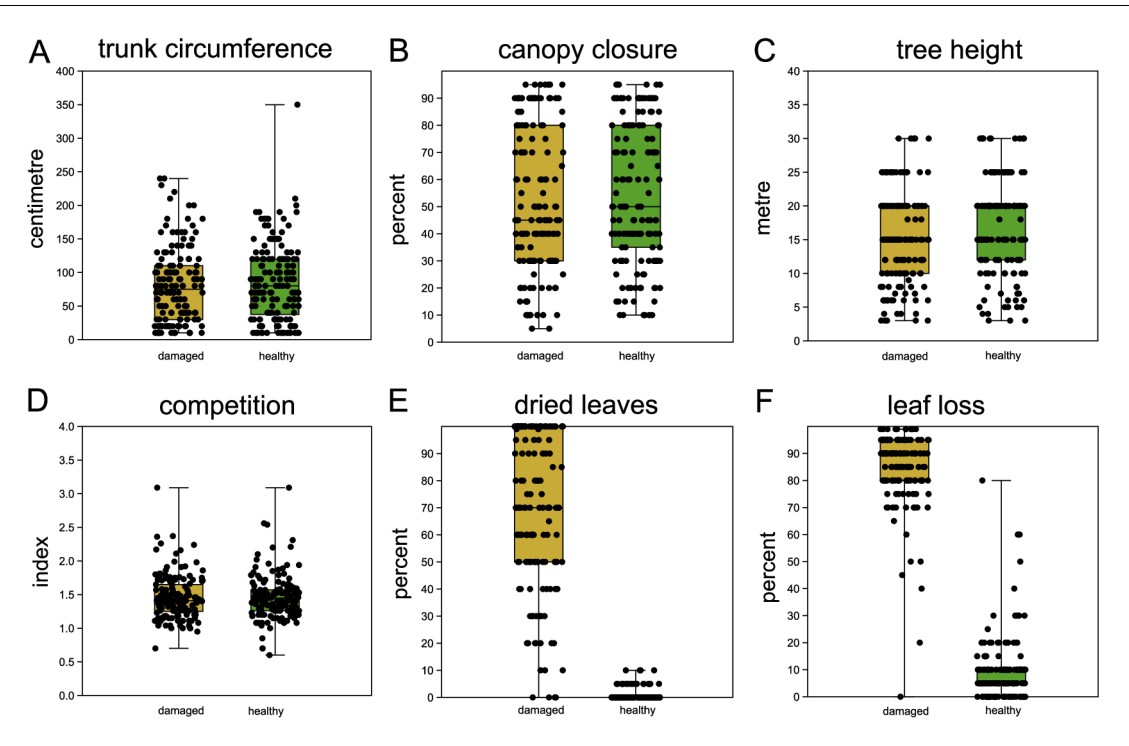

**Figure 2.** Comparison of sampled beech pairs. (**A**) Trunk circumference, (**B**) canopy closure, (**C**) tree height, (**D**) competition index, (**E**) dried leaves, and (**F**) leaf loss. Boxplots with indicated means, the boxes represent one standard deviation, the whiskers are the 95% confidence intervals. Damaged trees in ochre, healthy trees in green. Except for (**E**) and (**F**), the difference of means among damaged and healthy trees is insignificant between the groups.

The online version of this article includes the following figure supplement(s) for figure 2:

**Figure supplement 1.** Distribution of pairwise distances between the paired trees.

**Figure supplement 2.** Exemplary pictures of damaged and healthy beech tree pairs from several sampling sites.

---

population) did not tend to cluster together (*Figure 3—figure supplement 1B*). $F_{ST}$ estimates among pools for non-overlapping 1 kb windows were virtually identical among healthy/damaged pools within region as compared to between regions (*Figure 3—figure supplement 2*). Trees within a phenotypic class were genomically not more similar than between classes (*Figure 3—figure supplement 3*, ANOSIM R = 0.008, p=0.76, 9999 permutations).

Pool-GWAS analysis identified 106 SNPs significantly associated with the drought damage status using a Cochran–Mantel–Haenszel test on the two pairs of damaged and healthy pools after false discovery rate correction and a cutoff at $1 \times 10^{-2}$ (*Figure 3A*, *Figure 3—figure supplement 4*). Some of the 106 SNPs were in close physical proximity (<120 bp) and thus probably linked. Taking this into account, 80 independent genomic regions were associated with the drought damage status. None of the significantly differentiated SNP loci was mutually fixed; the observed allele frequency differences between healthy and damaged trees at associated loci ranged between 0.12 and 0.51 (*Figure 3B*).

## Associated genes and gene function

Of the 106 significant SNPs, 24 were found in 20 protein coding genes (*Table 1*). Forty-nine genes were the closest genes to the remaining 82 SNPs. For 61 of these genes, the best BLAST hit was with a tree, mainly from the Fagales genera *Quercus* and *Castanea* (*Table 1*, *Supplementary file 1C*). Among the 24 SNPs in genes, we observed 13 non-synonymous changes. In 11 of these changes, the alternate allele was associated with the damaged phenotype and only in two cases with the healthy phenotype. Three of the non-synonymous substitutions resulted in a stop codon. Of the remaining 10, 8 exchanges caused a major change in amino acid characteristics and thus

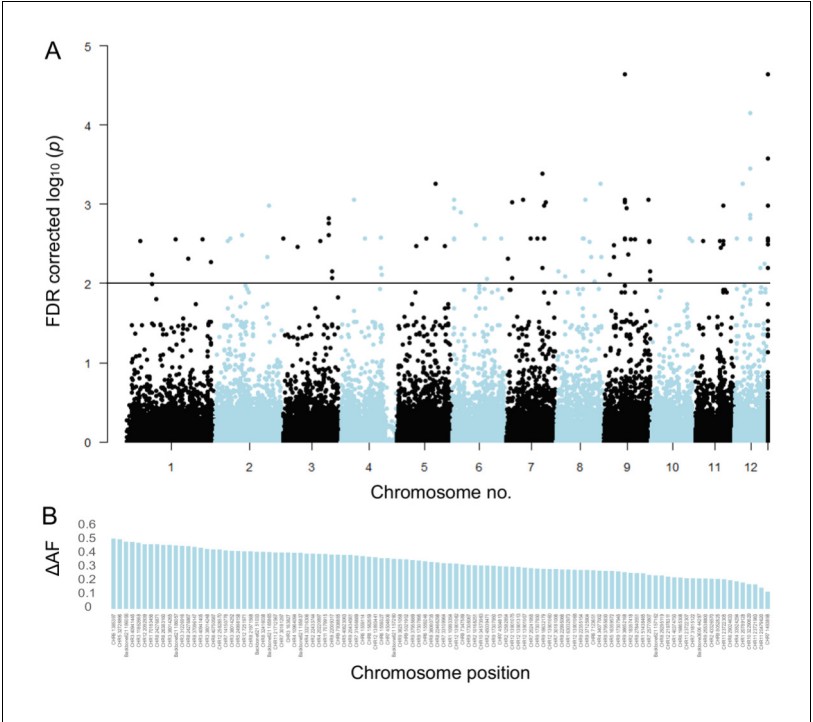

**Figure 3.** Single-nucleotide polymorphisms (SNPs) significantly associated with damaged phenotypes and combined results of SNP assay and discriminant analyses. (**A**) Manhattan plot of false discovery rate (FDR)-corrected –log10 probability values from Cochran–Mantel–Haenszel test. The black horizontal line indicates the chosen significance threshold. SNPs on different chromosomes alternate in colour (black and blue). (**B**) Mean allele frequency difference at significantly associated SNP loci between healthy and damaged phenotypes. The loci are ordered according to amount of change. (**C**) The centre of the figure depicts the genotypogram of the SNP assay. Each column represents one of 70 loci, each row one of 92 beech individuals. The scored genotypes are colour-coded, with red squares = homozygous reference allele, light blue = homozygous alternate allele, white = heterozygous SNP, and grey squares = locus could not be scored in the respective individual. The left bar indicates the observed phenotype for each tree individual with ochre rectangles for damaged and green for healthy trees. Below the genotypogram, the relative contribution of each locus to the predictive model of the discriminant analysis is indicated, ordered from high to low. On the right side, first the genotype model scores for each individual are given, with the according predicted phenotype (ochre = damaged; green = healthy).

The online version of this article includes the following figure supplement(s) for figure 3:

**Figure supplement 1.** Genome-wide linkage disequilibrium (LD) and principal component analysis on genome-wide single-nucleotide polymorphism (SNP) data.

**Figure supplement 2.** Genome-wide $F_{ST}$ distributions in 1 kb windows.

**Figure supplement 3.** Genomic similarity among individuals within and among phenotypic classes.

**Figure supplement 4.** Uncorrected Manhatten-plot.

---

probably in protein folding or function (**Table 1**). One gene, a PB1 domain-containing protein tyrosine kinase, contained four non-synonymous changes, suggesting that the allele version associated with the damaged phenotype lost its function (**Table 1**). From the 20 genes with significant SNPs, functional information could be obtained from the UniProt database for 14 (**Supplementary file 1C**). Of these, 10 genes were associated in previous studies with either environmental stress response (two) or specifically with drought stress response (eight; **Supplementary file 1C**). Of the 49 predicted genes closest to the remaining significant SNPs (**Table 1**), 16 could be reliably annotated (**Supplementary file 1B**). Twelve had been directly related to drought in previous studies, while three were previously associated with other environmental stress responses (**Supplementary file 1C**).

**Table 1.** Genes with significantly associated single-nucleotide polymorphisms (SNPs).

Given are the chromosome number (CHR), nucleotide position (position), the gene ID for *Fagus sylvatica* (gene), the UniProt ID of the closest match (UniProt ID), the name of the gene (name), the nucleotide base in the reference (ref DNA base), and the alternate base (alt DNA base), if applicable, the amino acid of the reference (ref AA) and the alternate base (non-synonymous change), functional change (effect), and the phenotype associated with the alternate base.

| CHR | Position | Gene | UniProt ID | Name | Ref DNA base | Alt DNA base | Ref AA | Non-synoynmous change | Effect | Phenotype associated with alternate base |
|---|---|---|---|---|---|---|---|---|---|---|
| 1 | 40374762 | 1 . g3851. t1 | None | | A | G | C | R | SH side chain > positive charge | Healthy |
| 10 | 32290645 | 10 . g3914. t1 | None | | C | T | F | - | - | |
| 11 | 20479628 | 11 . g2467. t1 | EXOS5_ORYSJ | Exosome complex exonuclease RRP46 homolog | T | C | T | A | Polar > hydrophobic | Damaged |
| 11 | 23722307 | 11 . g2832. t1 | PCN_ARATH | WD repeat-containing protein PCN | A | G | I | V | Hydrophobic > hydrophobic | Damaged |
| 12 | 13901034 | 12 . g1695. t1 | F4I5S1_ARATH | PB1 domain-containing protein tyrosine kinase | C | A | P | Q | Hydrophobic > polar | Damaged |
| | 13901063 | | | | C | T | Q | Stop | Termination | Damaged |
| | 13901082 | | | | A | T | H | L | Positive charge > hydrophobic | Damaged |
| | 13901094 | | | | T | A | I | N | Hydrophobic > polar | Damaged |
| 2 | 43326571 | 2 . g4736. t1 | None | | G | A | R | C | Positive charge > SH side chain | Damaged |
| 3 | 31226940 | 3 . g3590. t1 | None | | C | T | Q | Stop | Termination | Healthy |
| 4 | 34077017 | 4 . g3980. t1 | CKX1_ARATH | Cytokinin dehydrogenase 1 | C | A | G | C | No side chain > SH side chain | Damaged |
| 5 | 16359587 | 5 . g1807. t1 | GDI2_ARATH | Guanosine nucleotide diphosphate dissociation inhibitor 2 | T | C | P | - | - | |
| 6 | 19865311 | 6 . g2227. t1 | NDUS7_ARATH | NADH dehydrogenase (ubiquinone) iron-sulfur protein 7, mitochondrial | T | C | D | - | - | |
| 6 | 26383172 | 6 . g2921. t1 | None | | T | C | A | - | - | |
| 7 | 1493904 | 7 . g177. t1 | TLP10_ARATH | Tubby-like F-box protein 10 | C | T | G | - | - | |
| 7 | 20242023 | 7 . g2350. t1 | PRK4_ARATH | Pollen receptor-like kinase 4 | A | G | P | - | - | |
| 7 | 4504799 | 7 . g1655 | | | C | T | W | Stop | Termination | Damaged |
| 7 | 31456694 | 7 . g3617. t1 | LSH4_ARATH | Protein LIGHT-DEPENDENT SHORT HYPOCOTYLS 4 | G | T | R | - | - | |
| 7 | 33110000 | 7 . g3816. t1 | None | | G | A | L | - | - | |

*Table 1 continued on next page*

*Table 1 continued*

| CHR | Position | Gene | UniProt ID | Name | Ref DNA base | Alt DNA base | Ref AA | Non-synoynmous change | Effect | Phenotype associated with alternate base |
|---|---|---|---|---|---|---|---|---|---|---|
| 7 | 4504813 | 7 . g552. t1 | None | | G | A | G | - | - | |
| | 4504831 | | | | C | T | L | - | - | |
| 8 | 29295139 | 8 . g3494. t1 | VATC_ARATH | V-type proton ATPase subunit C | G | A | G | - | - | |
| 9 | 25538827 | 9 . g3080. t1 | PPA14_ARATH | Probable inactive purple acid phosphatase 14 | G | C | K | N | Positive charge > polar | Damaged |
| 9 | 37955715 | 9 . g4504. t1 | TBL33_ARATH | Protein trichome birefringence-like 33 | G | C | M | I | Hydrophobic > hydrophobic | Damaged |

## Genomic prediction

We furthermore set out to determine how many SNPs were needed to successfully predict the drought susceptibility of individual trees, that is, to develop a genotyping assay. All Pool-GWAS SNPs in addition to the top 20 individual resequencing SNPs were used to create an SNP combination to reach a genotyping success threshold of min 90%. After excluding loci due to technical reasons and filtering for genotyping success, 70 loci proved to be suitable for reliable genotyping with an SNP assay. We genotyped only individuals sampled in 2019 that were not used to identify the SNPs in the first place plus paired individuals sampled in August 2020. On average, each of the 95 individuals was successfully genotyped at 67.7 loci (96.7%). We coded the genotypes as *0* for homozygous reference allele, *1* for heterozygous, and *2* for the homozygous alternate allele, thus assuming a linear effect relationship. *Figure 3B* shows the genotypogram for the tested individuals.

We applied a non-parametric Machine Learning algorithm for simultaneous feature selection and clustering that was especially designed for small sample sizes (*Gerber et al., 2020*; *Horenko, 2020*). By selecting the 20 most informative SNPs, the method identified four different clusters. Forty-seven resistant trees were correctly assigned to cluster 1 and 2, while nine susceptible trees were falsely allocated there. Thirty-nine susceptible trees were correctly assigned in clusters 3 and 4, while three resistant trees also fell in these categories. In this way, 88% of the 98 trees could be correctly classified according to their observed phenotype.

## Discussion

Over the last two decades, increasing drought periods caused severe damage to European forests (*Schuldt et al., 2020*; *Etzold et al., 2019*; *Pretzsch et al., 2013*). Conifers seem to suffer the most, but deciduous trees were also strongly affected (*Schuldt et al., 2020*). Weather data from our study area from 1950 onwards suggested that the climatic conditions for beech trees in the area investigated changed dramatically during this period. Roughly estimating the tree age from their trunk circumference (*Bošeľa et al., 2014*), more than a third of the trees were already in place at the beginning of this period. About 60% were recruited prior to the acceleration of temperature change from the 1980s onwards. As a result, trees in the mountainous regions of the study area today experience climatic conditions comparable to those experienced by lowland trees in the 1950s, which in turn now experience a climate that used to be typical for regions much further South. Given the documented propensity of beech for local adaptation (*Gárate Escamilla et al., 2019*; *Pluess et al., 2016*; *Aranda et al., 2015*), including drought (*Bolte et al., 2016*), it is therefore conceivable that current conditions exceed the reaction norm of some previously locally well-adapted genotypes with detrimental consequences for their fitness. If the trend of an increasingly drier vegetation period persists, this will likely affect an even larger proportion of the currently growing beeches.

Evolutionary genomics will be indispensable to predict and manage the impact of global change on biodiversity (*Waldvogel et al., 2020*). As already shown for other partially managed (tree) species (*Stocks et al., 2019*), in particular pool-GWAS approaches (*Endler et al., 2016*) have proven to be useful in guiding conservation management.

Our strictly pairwise sampling design avoided many pitfalls of GWAS studies, arising, for example, from cryptic population structure and shared ancestry (*Hoban et al., 2016*; *Wellenreuther and Hansson, 2016*). Despite presented evidence from this and other studies (*Schuldt et al., 2020*) that the observed crown damages in large parts of Central Europe used for phenotyping here are directly or indirectly due to the severe drought years 2018 and 2019, we must acknowledge that we have no direct physiological proof that the trees surveyed here indeed suffered from drought stress. In addition, the observed diagnostic symptoms are not specific to drought stress. Nevertheless, an unknown independent stressor would have needed to accidentally co-occur spatially and temporally with the drought. The phenotypical drought response of individual trees may also be influenced by microspatial variation (*Carrière et al., 2020*). In the present study, however, the mean distance between sampled paired trees of about 5 m assured that their roots systems largely overlapped. Thus, environmental variation in soil quality, rooting depth, water availability, or other factors should have been minimal. Please note that any phenotypical misclassification due to such microspatial differences would have rather dissimulated the genotypic differences found in GWAS than enhanced them artificially.

As expected from previous studies (*Rajendra et al., 2014*), we found no population structure among the sampling sites. Applying relatively strict significance thresholds, we found systematic genomic differences between the healthy and damaged trees. In all cases, these differences were quantitative and not categorical, that is, we found allele frequency changes but no fixed SNPs between phenotypes. Significant SNPs were mostly not clustered – we found on average 1.4 selected SNPs in a particular genomic region. These findings were in line with the observed very short average LD in *F. sylvatica*, indicating that polymorphisms associated with the two phenotypes were likely old-standing genetic variation (*Harris and Nielsen, 2013*). Moreover, such SNPs are mostly detached from the background in which they arose and they are therefore often the actual causal variants or at least in very close proximity (*Schaid et al., 2018*). This observation is underlined by the high proportion of non-synonymous significant SNPs within genes, which in most cases caused substitution to an amino acid with different properties or even premature termination. Such deviant variants with likely substantial functional or conformational changes in the resulting proteins may be selectively neutral or nearly neutral under ancestral benign conditions, but may become selectively relevant under changing conditions (*Paaby and Rockman, 2014*). Interestingly, most of the allelic variants associated with a healthy phenotype were also the variants in the reference genome. This might be due to the choice of the *F. sylvatica* individual from which the reference genome was gained (*Mishra et al., 2018*). This more than 300-year-old individual is standing at a particularly dry site on a rocky outcrop on the rim of a scarp where precipitation swiftly runs off. Trees at such sites were likely selected for drought tolerance.

Even though the area sampled for this study was limited relative to the species distribution range, it comprised its core area. In addition, the climatic variation covered by the sampling sites for this study is representative for large parts of the species range (*Baumbach et al., 2019*). The relatively limited population structure over large parts of the species range (*Magri et al., 2006*), together with the propensity for long-range gene flow (*Belmonte et al., 2008*), suggested that the genomic variation responsible for drought tolerance identified here is widely distributed (*Lander et al., 2021*). Nevertheless, an assessment of the geographic distribution of the drought-related genomic variants over the entire distribution range would yield general insight into the species-wide architecture of this important trait.

None of the genes found here was involved in a transcriptomic study on drought response in beech saplings (*Müller et al., 2017*). However, most of the reliably annotated genes with or close to SNP loci significantly associated with drought phenotypes had putative homologs in other plant species previously shown to be involved in drought or different environmental stress response (for citations, see *Supplementary file 1C, D*). For the remaining, not annotated genes, it remained unclear whether they had really never been associated with drought before, or whether we were just unable to make this link due to the lack of (ecological) annotation and standardised reporting (*Waldvogel et al., 2021*). The involvement of in total 67 genes, together with the relatively flat

effect size distribution, suggested that drought resistance in *F. sylvatica* is a moderately polygenic trait, which should respond well to artificial breeding attempts and natural selection. However, given the relatively strict threshold criteria, it is likely that more yet undetected loci contribute to the respective phenotypes. The low LD in beech predicts that an adaptation to drought will not compromise genome-wide genetic diversity and thus adaptation potential to other stressors. We achieved a high level of accuracy using genomic data to statistically predict the drought phenotype from individuals not used to identify drought-associated SNP loci. We used a non-parametric machine learning algorithm that has been shown to produce robust results, especially for small sample sizes (*Horenko, 2020*). Please note that the method is not trying to causally and quantitatively explain phenotypic differences, but uses statistical associations for prediction. The analysis confirmed that we mainly identified alleles widespread throughout the sampled range and not locally specific. Besides, we confirmed a considerable level of genetic variation in the sampled regions. The observation that trees with the highest predictive values showed no loss of heterozygosity indicated that there is still adaptive potential for drought adaptation in the species (*Gienapp et al., 2017*). With the SNP assay, we therefore created a tool that can (i) support the choice of seed trees for reforestations, (ii) provide decision guidance for selective logging, and (iii) monitor whether natural selection on this quantitative trait is already acting in the species. The current study can also serve as a starting point for molecular and physiological research on how the identified loci or variants may, alone or in concert, confer resilience or tolerance to a range of drought stress symptoms.

## Materials and methods

### Sampling and phenotyping

In August/early September 2019, we sampled leaf tissue of 402 *F. sylvatica* trees from 32 locations in Hessen/Germany (set 1, *Figure 1*), of which 300 were used for the (pool)GWAS analysis. 43, plus additional 53 trees which were sampled in August 2020, additional 52 trees from four sites were sampled (set 2, *Figure 1*) which made up the confirmation set. The coordinates and characteristics of each site can be found in *Supplementary file 1A*. The sampling was performed in a strictly pairwise design. The pairs consisted of one tree with heavy damage of the crown (lost or rolled up, dried leaves) and one with an unaffected crown, respectively. This categorisation into least and most damaged trees was taken compared to the other trees in the respective forest patch. The pairs were a priori chosen such that the two trees were (i) mutually the closest neighbours with contrasting damage status (i.e. no other tree in the direct sight line), (ii) free from apparent mechanical damage, fungal infestations, or other signs of illness, similar (iii) in tree height, (iv) trunk circumference, (v) light availability, and (vi) canopy closure. In addition, each pair was situated at least 30 m from the closest forest edge. For each tree of the chosen pairs, we recorded the exact position, distance to the pair member and the estimated tree height (in 1 m increments), measured the trunk circumference at 150 cm height above the ground (in 10 cm increments), and estimated the leaf loss of the crown and the proportion of dried leaves (in 5% increments). We also recorded the estimated distance (in 1 m resolution) and the specific identity of the two closest neighbour trees for each pair member and calculated a competition index $C$ as follows: $C = S_1/D_1 + S_2/D_2$, where $S_1$ and $S_2$ are the trunk diameter at 150 cm and $D_1$ and $D_2$ the distances of the nearest and second nearest neighbour tree of the same size or larger than the focal tree. Photographs from the crown and the trunk were taken from the trees sampled in 2019.

From each tree, we sampled 5–10 fully developed leaves from low branches. The leaves sampled from each tree were placed in paper bags. After returning from the field, they were dried at 50°C for 30–90 min and then kept on salt until they could be stored at 80°C.

### Climate and remote sensing data

Monthly daily mean minimum and maximum temperature values and precipitation data were obtained for the 1 1 km grid cells harbouring the sampling sites for the period between 1950 and 2019. Data on the accumulated potential evapotranspiration during the growth season was obtained for the same grid cells. The data is publicly available from https://opendata.dwd.de/climate_environment/CDC/grids_germany/monthly/.

LAI data for the above grid cells was obtained from Copernicus (http://www.copernicus.eu) for the period 2014–2019, considering only the month of August. To see whether drought conditions influenced leaf coverage of the woods at the sampling sites, we calculated the relative annual deviation of LAI from the 2014 value. We correlated it to the relative deviation of the cumulated potential evatransporation over the growth season from 2014. The year 2014 was used as a baseline because of the significant drought increase since then (*Büntgen et al., 2021*). Please note that the absolute level of LAI depends on the wood coverage, vegetation density, and species composition of each plot. Changes in LAI are thus not exclusively due to drought damage in beech.

## DNA extraction, construction of GWAS pools, and sequencing

DNA was extracted from 12.5 mm$^2$ of a single leaf from each tree following the NucleoMag Plant Kit (Macherey Nagel, Düren, Germany) protocol. We setup four DNA pools for Pool-GWAS by pooling equal amounts of DNA from each individual: damaged individuals from the Southern part (dSouth), healthy individuals from the South (hSouth), damaged North (dNorth), and healthy North (hNorth). The Southern pools consisted of 100 individuals each, the Northern pools of 50 individuals each. The pools were sent to Novogene (Cambridge, UK) for library construction and 150 bp paired end sequencing with 350 bp insert size with 25 Gb data for the Northern and 38 Gb data for the Southern samples. The 100 individuals used to construct the Northern pools were also individually resequenced. The exact composition of the genomic pools can be found in *Supplementary file 1A*. All sequence information can be found on the European Nucleotide Archive (ENA) under project accession number *PRJEB24056*.

## Reference genome improvement

We used an improved version of the recently published reference genome for the European beech (*Mishra et al., 2018*). Contiguity was improved to chromosome level using Hi-C reads with the help of the allhic software after excluding the probable organelle backbones from the earlier assembly that was generated from the Illumina-corrected PacBio reads using Canu assembler (*Mishra et al., 2021*) accession no. PRJNA450822.

## Mapping and variant calling

Reads of pools and individual resequencing were trimmed using the wrapper tool autotrim v0.6.1 (*Waldvogel et al., 2018*) that integrates trimmomatic (*Bolger et al., 2014*) for trimming and fastQC (*Andrews, 2010*) for quality control. The trimmed reads were then mapped on the latest chromosome-level build of the *F. sylvatica* genome using the BWA mem algorithm v.0.7.17 (*Li and Durbin, 2009*). Low-quality reads were subsequently filtered and SNPs were initially called using samtools v.1.10 (*Li et al., 2009*). A PCA was conducted on unlinked SNPs using the R package Factoextra v.1.0.7 (*Kassambara and Mundt, 2017*).

## Pool GWAS and PLINK

The PoPoolation pipeline 2_2012 (*Kofler et al., 2011a*; *Kofler et al., 2011b*) was used to call SNPs and remove indels from the four pools. Allele frequencies for all SNPs with a coverage between 15 and 100   with a minimum allele count of 3 were estimated with the R library PoolSeq v. 0.35 (*Taus et al., 2017*).

The statistical test to detect significant allele frequency differences among damaged and healthy trees was the Cochran–Mantel–Haenszel test. With this test, a 2   2 table was created for each variable position and region with two phenotypes (healthy and damaged). The read counts of each allele for each phenotype were treated as the dependent variables. We controlled for false discovery rate using the Benjamini–Hochberg correction in R package *p.adjust*.

For the individual resequencing data, we followed the GATK-pipeline 4.1.3.0 (*DePristo et al., 2011*). In short, Picard tools v.2.20.8 was used to mark duplicates. GVCF files were created with HaplotypeCaller and genotyped with GEnotypeGVCFs. Since we did not have a standard SNP set, we hard-filtered SNPs with VariantFiltration QD < 2.0, MQ < 50.0, MQRankSum < 12.5, ReadPosRankSum < 8.0, FS > 80.0, SOR > 4.0, and QUAL < 10.0. This conservative SNP set was used for base recalibration before running the HaplotypeCaller pipeline a second round. Finally, the genotyped vcf-files were filtered using vcftools with `–maf` 0.03 `–max-missing` 0.9 `–minQ` 25 `–min-`

`meanDP` 10 `–max–meanDP` 50 `–minDP` 10 `–maxDP` 50. The detailed pipeline can be found in *Supplementary file 1G*.

To conduct the GWAS association on the above-generated SNP set with phenotypes being either damaged or healthy and to generate a PCA on the SNP positions of the individually resequenced trees, we used PLINK 1.9 (*Purcell et al., 2007*). The detailed workflow can be found in *Supplementary file 1G*. We calculated a non-parametric ANOSIM on an inter-individual Euclidean distance matrix based on the first 10 principal components to infer whether the trees within phenotype groups are overall genetically more similar than within groups (9999 permutations; *Hammer et al., 2001*).

## Inference of linkage disequilibrium

The expected length of segregating haplotypes in a species depends on the recombination rate and their age. The former can be approximated by an estimate of LD. To determine LD decay based on individually resequenced data, we used the software LDkit v 1.0.0 (*Tang et al., 2020*), in 1 kb and 100 kb windows.

## Identification substitution type and gene function

We inferred whether significantly differentiated SNPs within genes lead to a (non-) synonymous amino acid substitution using tbg-tools v0.2 (https://github.com/Croxa/tbg-tools; *Schoennenbeck et al., 2021*). The protein sequences of the identified genes were used in a blastp search against all non-redundant GenBank CDS translations, PDB, SwissProt, PIR, PRF to infer potential gene functions. For each search, only the single BLAST- top hit was considered.

## Selection of SNP loci for SNPtype assay design

For the design of SNPtype assays, we used the web-based D3 assay design tool (Fluidigm Corp.). We aimed in first preference for the most significant SNPs of each genomic region identified by Pool-GWAS (80 loci). If this was technically impossible and the region harboured more than a single significant SNP, we opted for the second most significant SNP and so forth. This resulted finally in 76 suitable loci. The remaining 20 loci were recruited from the 20 most significant SNPs of the PLINK analysis that were not scored in the Pool-GWAS.

## SNP genotyping procedure

For validation of drought susceptibility-associated SNPs, we conducted SNP genotyping on 96.96 Dynamic Arrays (Fluidigm) with integrated fluidic circuits (*Wang et al., 2009*) (N = 96) to validate the effectiveness of the identified SNPs in discriminating healthy from damaged trees. Prior to genotyping PCR, DNA extracts were normalised to approximately 5–10 ng/µl. They underwent a pre-amplification PCR (Specific Target Amplification [STA]) according to the manufacturer's protocol to enrich target loci. PCR products were diluted 1:10 with DNA suspension buffer (TEKnova, PN T0221) before further use. Genotyping was performed according to the manufacturer's recommendations. Four additional PCR cycles were added to accommodate for samples of lower quality or including inhibitors (*von Thaden et al., 2020*). Fluorescent data were measured using the EP1 (Fluidigm) and analysed with the SNP Genotyping Analysis Software version 4.1.2 (Fluidigm). The automated scoring of the scatter plots was checked visually and, if applicable, manually corrected.

## Genomic prediction

To predict drought susceptibility from genotype data, we used a non-parametric entropy-based Scalable Probabilistic Analysis framework (eSPA*, *Vecchi et al., 2022*). This method allows simultaneous solution of feature selection and clustering problems, meaning that does not rely on a particular choice of user-defined parameters and has been shown to produce more robust results than most other machine learning classification algorithms, especially for small sample sizes (*Gerber et al., 2020*; *Horenko, 2020*). For the optimization process, eSPA* calculates a probability for each beech to belong to one of k clusters, with each cluster representing a specific combination of genotypes at SNP sites. Each cluster in turn has a specific probability of including either drought-resistant, or drought sensitive beeches. If a beech is assigned to e.g. cluster 1 (pattern 1) based on its specific SNP profile this means that this beech has at least a 79% Likelihood to be drought

resistant. All beech trees assigned to a certain cluster are also given a probability indicating how "well" this beech fits to this cluster. This is estimated using a distance function that calculates the goodness of fit to the cluster. We applied a cross-validation approach with 100 independent runs of 75% of the samples as training and 25% of the samples as test-set. By iterating this process over several number of clusters, the optimal solution is found.

## Acknowledgements

There was no external funding for this study. We thank the SBiK-F Data- and Modelling Centre (Aidin Niamir) for support in obtaining climate and remote sensing data. CC acknowledges support from the Emergent AI Center funded by the Carl-Zeiss-Stiftung.

## Additional information

### Funding

No external funding was received for this work.

### Author contributions

Markus Pfenninger, Conceptualization, Resources, Data curation, Formal analysis, Supervision, Funding acquisition, Investigation, Visualization, Writing - original draft, Project administration; Friederike Reuss, Supervision, Investigation, Writing - review and editing; Angelika Kiebler, Investigation, Writing - review and editing; Philipp Schönnenbeck, Software, Formal analysis, Writing - review and editing; Cosima Caliendo, Susanne Gerber, Formal analysis, Writing - review and editing; Berardino Cocchiararo, Investigation, Methodology, Writing - review and editing; Sabrina Reuter, Formal analysis, Investigation, Writing - review and editing; Nico Blüthgen, Conceptualization, Supervision, Writing - review and editing; Karsten Mody, Conceptualization, Investigation, Writing - review and editing; Bagdevi Mishra, Marco Thines, Resources, Writing - review and editing; Miklós Bálint, Conceptualization, Formal analysis, Writing - review and editing; Barbara Feldmeyer, Data curation, Formal analysis, Writing - review and editing

### Author ORCIDs

Markus Pfenninger (iD) https://orcid.org/0000-0002-1547-7245

### Decision letter and Author response

Decision letter https://doi.org/10.7554/eLife.65532.sa1
Author response https://doi.org/10.7554/eLife.65532.sa2

## Additional files

### Supplementary files

• Source code 1. eSPA* input-file for MatLab.

• Supplementary file 1. Tables with detailed information. (**A**) Sampled *Fagus sylvatica* individuals. (**B**) Results from Mann–Whitney U-test on difference between damaged and healthy trees for various parameters. (**C**) List of genes with significant single-nucleotide polymorphisms (SNPs). (**D**) List of genes closest to significant SNPs. (**E**) List of 20 most informative SNPs as selected by the entropy-based Scalable Probabilistic Analysis (eSPA) allowing for 85% correct classification. (**F**) Software pipeline and commands used for PoolSeq analysis. (**G**) Workflow individual reseq GWAS.

• Transparent reporting form

### Data availability

Sequencing data have been deposited at ENA under project code PRJEB41889. The genome assembly including the annotation is available under the accession PRJNA450822. eSPA* input-file for MatLab, containing both source data and source code is available as Source code 1.

The following dataset was generated:

| Author(s) | Year | Dataset title | Dataset URL | Database and Identifier |
|---|---|---|---|---|
| Pfenninger M, Feldmeyer B | 2020 | Genomic basis of drought resistance in Fagus sylvatica by PoolGWAS | https://www.ebi.ac.uk/ena/browser/view/PRJEB41889 | European Nucleotide Archive, PRJEB41889 |

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
