## [Decision Letter]

**Acceptance summary:**

This study is a pioneering example of something like "personalized medicine" in the sense of diagnosis, applied to forests. The authors study a tree species that forms a large proportion of European forests, using real trees in medium- to old-growth forests. Using a paired sampling design combined with ecological and climatological observations, they are able to diagnose variants indicative of stress resistance or tolerance which vary among neighboring trees. In doing so the authors present new quantitative evidence that European beech forests are strongly affected by climate change, and that the genome of beech harbors resistant genes which may be broadly distributed in trees across its range.

**Decision letter after peer review:**

Thank you for submitting your article "Genomic basis of drought resistance in Fagus sylvatica" for consideration by *eLife*. Your article has been reviewed by 3 peer reviewers, including Meredith C Schuman as the Reviewing and Senior Editor and Reviewer #3. The following individual involved in review of your submission has agreed to reveal their identity: Lucienne de Witte (Reviewer #1).

The reviewers have discussed their reviews with one another, and the Reviewing Editor has drafted this evaluation to help you prepare a revised submission.

Summary:

The reviewers agree that the study is of broad interest and that the genetic work represents a substantial advance on existing tools in European beech, an economically and ecologically important forest species with an uncertain future under global change. The reviewers value the case-control design using trees in standing populations and taking advantage of existing environmental data. There are questions about many important details both regarding tree selection and phenotyping, and also the genetic analysis, which are missing from the manuscript and need to be addressed. The results provide a valuable basis for the development of hypotheses regarding the adaptive potential within beech populations facing global change, but many additional steps would be required to articulate and test specific hypotheses emerging from these results. Thus we would consider a revised version of this manuscript, addressing these essential revisions and considering the other points raised, for publication as a Tools and Resources article formatted as a Short Report, to accompany the sequencing dataset that the authors have already deposited at ENA, including an update of the beech genome (also at ENA/beechgenome.net) corresponding to the revisions made in the course of this study. The revision would need to include a full description (including specification of all data, software and parameter settings used) of how the assembly was improved, as well as details of the annotation (both genes and repeats).

Essential Revisions:

This is a list of revisions considered essential, referring to more detailed information from the individual reviews where relevant. The individual reviews are also appended in full for your information, including public reviews and a public summary designed to accompany your preprint, and recommendations for the authors from each reviewer.

1) The authors state that they selected trees which were the nearest neighbors showing distinct phenotypes (apparently unaffected canopies versus canopies with dead and curled, dried leaves) and not displaying any apparent indications of insect infestation or disease which would cause such differences. This stressed versus healthy classification used by the authors is not specific to drought. To the extent that their identified genetic targets could be annotated, most of these are also generally stress responsive and not specific to drought stress. Drought is a complex stressor and it is certainly plausible that drought could be the main cause of different stress status in this study, but the authors do not present evidence to support this other than the general knowledge that beech forests suffered under the European 2018 summer drought and other recent droughts.

– More direct indicators of water stress would provide better support for their classification, if any such data exist. Please refer to comments and specific suggestions from Reviewer #1 – Recommendations for authors point 1, and Reviewer #3 – Public review, "major weaknesses" section.

– Information on the relative health status of the chosen trees prior to the 2018 extreme drought could also be valuable, and perhaps the different climatic conditions of their study populations could be better leveraged to support the importance of drought in explaining different apparent stress status in the pairs of trees they sampled.

– To the extent that the authors cannot provide data on study trees which is more specific to drought stress, they must acknowledge that their stress classification cannot be clearly attributed to drought susceptibility of the trees, and may incorporate other (multiple) stress factors.

2) Beyond the concerns regarding interpretation of tree stress status, sufficient information must be provided regarding the trees studied.

– The authors should provide evidence to support their classification of trees as healthy versus stressed (such as images – even example photographs, but better systematic photographs or aerial imagery of pairs; and/or supporting measurements; field rankings of e.g. crown transparency or apparent stress based on specific criteria).

– The authors should provide at least a rough measurement of how far pairs of trees were allowed to be apart (e.g. minimum and maximum distance in m) and roughly how many other trees were allowed to stand in between (to what extent can these trees be considered neighbors). Please refer to Reviewer #3 – Public review, "major weaknesses" section.

– The authors must either provide information on known or expected within-site variation of water availability and other relevant factors such as soil quality, possible rooting depth, and differences in biotic interactions which could have affected drought status or responses, on the scale of distance between their selected pairs; or at a minimum, they should discuss possible environmental variation on this scale based on literature such as that suggested by Reviewer #1.

– There is conflicting information regarding numbers of individuals sampled, and included in each group, which must be resolved. Please refer to comments from Reviewer #2 under Recommendations for authors – Major suggestions – Number of individuals used.

3) There is also important information missing regarding sample and data analysis, and this must be provided, and concerns regarding non-standard methods addressed. Please see detailed comments from Reviewer #2 (Recommendations for authors – Major suggestions) regarding the following:

– Insufficient information is provided regarding the improvement of the reference genome.

– The authors used unevenly sized pools for poolGWAS and equal membership of pooled samples can also not be guaranteed based on the information provided in the Materials and methods.

– Insufficient information is provided regarding how GWAS was conducted.

– It is not clear which indicators of statistical significance are reported and how these are to be interpreted.

– The reported very low LD may be an artifact of interpreting LD from poolGWAS data, and should be assessed using the individual genomes which were sequenced at high coverage.

– It is not clear how group membership prediction was conducted and how the very high success rate is to be interpreted (GP analysis).

– The functional interpretation of SNPs is complicated by some apparent contradictions in genome locations and gene models, some of which may be resolved by more information from the improved genome assembly.

– Insufficient information is provided regarding software versions and parameters used, and also reasoning for choosing some non-standard software tools.

– Additionally, from Reviewer # 3, Recommendations for authors: The leaf handling protocol is unusual and it is not very precisely reported: collected fresh from field, 50 deg C in lab 30-90 min, then to salt until storage at -80. It seems like the DNA would be better preserved if leaves were harvested directly to silica gel and not heated as is commonly done (in which case storage at -80 deg C can be done but is generally not required).

4) One important hypothesis suggested by this study is that beech populations locally embody sufficient variation in drought-responsive genes to adapt to increased drought severity and frequency. However, this cannot be supported as a conclusion of the study. The authors should re-state this as a hypothesis and propose work required to test this hypothesis rigorously. As noted by Reviewer 1, the candidate genes revealed often do not overlap with those reported from other studies, including studies in beech, and as noted by Reviewer 3, the geographic limitations of the study preclude conclusions regarding to what extent the study populations embody the potential of members of the species to adapt to drought. Furthermore, there is insufficient phenotypic data to link the identified loci to specific response curves (e.g. change in ability to tolerate or recover from a given severity of water deficit and associated stresses such as salinity and heat). The ability to predict membership in a group (healthy versus stressed) by genotype demonstrates the consistency of associations identified within this dataset, but it does not indicate that the study populations are resilient to predicted future increases in drought frequency and severity. Again, rather, these results suggest hypotheses about how specific loci may, alone or in concert, confer resilience or tolerance to a range of drought stress, which could be better articulated based on knowledge in the beech system and gene annotation; and the authors could suggest follow-on studies, increasing the value of their own study as a resource.

*Reviewer #1 (Recommendations for the authors):*

1. The authors study the genomic basis of drought susceptibility and found systematic genomic differences. However, selection criteria of damaged trees are unclear because trait variables used for phenotype differentiation seem not consistent with protocols from internationally recognized monitoring networks.

In the first paragraph of the introduction (lines 55-58 and 68-74), it is stated that drought damage in beech includes rolled leaves and small leaves. But these two symptoms rather are resistance/stress symptoms, see also Wohlgemuth et al. 2020 (Schweiz Z Forestwes 171). Beech trees roll their leaves and produce smaller leaves to avoid to much evapotranspiration. The literature cited, Paar and Dammann 2019, state stress symptoms (Trockenstresssymptome) versus damage, the later defined as decreased growth, increased crown transparency and increased mortality. These definitions are according to standardized monitoring protocols used by international networks such as ICP forests (ICP forest manual). Why were these variables not included for tree selection?

In the Results chapter, lines 113-118, the damage status was assigned according to the quantity of dried leaves and leaf loss (Figure 2 E-F, Suppl. Table 2 is referred to but that table does not include any info on these variables, only the results from the Mann-Whitney U-test on differences between damaged and healthy trees). What do dried leaves and leaf loss mean? These are not usual terms or variables used to describe or name damage in beech trees in standardized monitoring. Do they translate to leaf discoloration and crown transparency?

In the Materials section (lines 237-254) you state again differently named traits: "heavy drought damage of the crown (lost or rolled up, dried leaves) (…) leaf loss of the crown and the proportion of dried leaves (in 5% steps)."

On lines 75-86 you use the terms „drought susceptibility" versus „resistant" trees which is good. The wording is more concise in this paragraph than first paragraph of the introduction.

Attention should be paid also to the damages recorded during monitoring: crown transparency, discoloration, decrease in growth and increase in mortality cannot be explained only by drought stress. Modelling analysis of long-term data for example shows that drought, next to other reasons, is a main reason for the actual damages observed in central European beech forests (Braun et al. 2020, Schweiz Z Forstwes 171). Observed damages in beech trees are not the consequence only of drought (temperature increase and precipitation decrease), but also of storm events (root damages!) and environmental pollution (including ozone and nitrogen deposition impact soil quality and growth of roots and associated microbiome), and because these factors also interact with drought. These factors should at least be considered and discussed (see also Pflug et al. 2018).

*Reviewer #2 (Recommendations for the authors):*

This is an interesting study on the genomic basis of drought resistance in an important forest tree species. However, I have concerns about some of the analyses and there are several aspects of the paper where there is a lack of detail or where further clarification is needed. I outline these points below.

Number of individuals used

I am confused about the number of individuals included in some of the analyses because there are conflicts between the details provided in different parts of the manuscript. For example, Lines 76-77 of the introduction suggest that more than 400 individuals were used for the pool-GWAS ("more than 200 neighbouring pairs of trees"), but Line 120 states that "300 beech trees" were used for the pool-seq. Lines 122-123 say that there were 100 individuals in each of the South pools, but, Table S1 indicates there are 135 trees in each South pool. On Lines 127-128 it states "All 100 individuals from the North population" were individually re-sequenced, but the Materials and Methods say 102 individuals were re-sequenced (Line 271) and Table S1 indicates that it was 95 individuals. On Lines 238-239 it says 53 trees were sampled for "set 2", but Table S1 indicates there are 80 individuals in set 2. There are various other similar examples, which altogether create a confusing picture. The Authors need to double-check all such details and correct them as necessary.

Reference genome

More details about the methods used for improving the reference genome are needed. For example, please provide more details of how the Hi-C reads were generated (i.e. source of DNA, extraction method, sequencing coverage) and add details of the version number and parameter settings used for the allhic software. Also, was the improved assembly reannotated? If so, how? If the original annotation was transferred to the new assembly, this should also be explained.

GWAS

The guidance for popoolation (https://sourceforge.net/p/popoolation2/wiki/Manual/) states that, for the experimental design, the most important thing is "the amount of DNA per individual in a single pool should be constant". In the Materials and Methods section that deals with DNA extraction (starting on Line 264), there is no mention of quantification of DNA being done to ensure that the amount of DNA contributed to a pool by each individual is equal. Please can the Authors provide details of how they quantified DNA from each individual prior to pooling to ensure they contributed equally? The guidance also states that, "the number of individuals per pool should be similar". However, the Southern pools are much larger than the Northern pools (more than double the size according to the information in Table S1). Could the Authors comment on why they used such unevenly sized pools and what impact this might have had on the results of the GWAS?

With regard to the potential impact of population stratification, which could create spurious associations in the GWAS, the Authors state on Lines 201-202 "As expected from previous studies, we found no population structure among the sampling sites". However, I cannot see anywhere an analysis of potential structure between all populations sampled for the main GWAS analysis (i.e. the pool-GWAS). Could the Authors comment on this further please?

Also, in relation to the control for FDR, the method used by the Authors (the R package qvalue) does not calculate "corrected" p-values. Normally for GWAS, false discovery rates are used to set an appropriate p-value significance threshold to control for the number of false discoveries, rather than to correct p-values as with multiple comparison correction. I.e. a threshold can be set under which the number of false positives is deemed to be acceptable. In Figure 4 the Authors present "corrected" -log10 p-values and indicate a p-value significance threshold. I think the values in this plot are actually the q-value estimates that are generated by the qvalue package. I suggest the Authors present the original p-values from the association analysis, but indicate a significance threshold depending on the number of false positives they are willing to allow within the set of "significant" SNPs. If this changes which SNPs are considered to be significant, then clearly any subsequent analyses and results that are presented should account for this fact.

Furthermore, for the GWAS analysis using data from the re-sequenced individuals (mentioned in Lines 303-308), please provide full details for what was actually done, including any additional filtering of SNPs done in PLINK (e.g. LD pruning, which is mentioned in the legend for Figure 3, but not in the Materials and Methods) and which statistical test for association was performed. Moreover, there don't seem to be any details for this GWAS analysis in the Results section, although it is mentioned on Lines 326-327 of the Materials and Methods that some of the significant SNPs were from this GWAS were included in the SNP array.

LD analysis

The Authors estimated LD in beech on the basis of a subset of the pool-seq data. This suggests a very short size of linkage block on average (c.120bp), which led the Authors to conclude that most of the significant SNPs they detected are "the actual causal variants" as they are unlikely to be linked to other SNPs. However, it has been suggested that the estimates of LD obtained from analysis of pool-seq data are not very accurate (Schlötterer et al., 2014; doi:10.1038/nrg3803). The finding that SNPs that are further than c. 120bp apart are likely to be unlinked is surprising to me given the length of linkage blocks that have been detected in some other tree species. As the Authors have whole genome sequence data for individuals from the Northern pools, I would recommend that they also use these data to estimate LD to see if this produces a result that is congruent with that based on the pool-seq data and supports their assertion that they have likely identified causative SNPs.

GP analysis

The details of the GP analysis need further clarification. In particular, I am confused by the reported level of accuracy of GP with the set of 70 variants. The Authors say "Linear discriminant analysis (LDA) correctly predicted the observed phenotype from the genotype in 91 of 92 cases (98.9%)". However, in the Materials and Methods it is reported that 80% of individuals were used in training and only 15 individuals were used in the test set. Can the Authors please make clear what they actually did to test the accuracy of the LDA analysis in discriminating trees with the different phenotypes? If only 15 trees were included in the test set, then accuracy should be reported based on the results of these. Also, was any replication done to see if results differ with different subsamples of 15 individuals (i.e. using multiple random samples of 15 individuals from the full set)? If not, then I think this would be worth considering. Alternatively, have the Authors considered also using some of the pool-seq data to train a GP model? This could be used to test the accuracy of prediction for the individually sequenced trees, which would provide a larger number of trees for testing. Further, the details of the number of individuals used for the GP analysis and numbers and percentages of individuals used for training and testing need to be double-checked (lines 342-346), as there seem to be some inaccuracies here.

Functional significance of SNPs

tbg-tools was used to check whether significant SNPs from the GWAS cause non-synonymous substitutions in non-coding regions. Could the Authors explain why they chose to use this tool (which is described on its GitHub page as "not in its finished version and there are surely still some bugs to be found") rather than more established software such as SnpEff (https://pcingola.github.io/SnpEff/)? Also, even if the software correctly predicts the impacts of the variants, these predictions are dependant on the accuracy of the genome annotation. As noted above, there is a lack of detail about the annotation for the improved reference genome (and also no indication of the availability of the annotation file). Could the Authors comment on whether they manually checked the gene models for those cases where a significant SNP fell within a gene (i.e. results in Table 1) to see if there was any evidence for gene model errors that could impact the results? For example, two variants are reported to be within gene model 7.g2350.t1, one of which causes a premature stop codon. However, the position of these variants suggests they are >15Mb apart; assuming the positions have been correctly reported, I would be surprised if this gene model is correct, especially as the best matching gene from *A. thaliana* is only c. 3kb. Also, in the case of 12.g1695.t1, four variants are reported with a c. 60bp region of the gene, including one inducing a premature stop codon in individuals with the alternate allele. This leads the Authors to suggest that the gene "lost its function" in individuals that carry the non-reference version of the allele. However, another possible cause of such a pattern would be incorrect specification of intron/exon boundaries in the gene model. Have the Authors double-checked this to confirm that the variants are actually within a protein-coding region? Even if no evidence of gene model errors is found, it cannot be concluded from the sequence data alone that the gene will be non-functional, as even with a premature stop codon a truncated product with some functionality could be formed. So, I suggest the Authors need to be a bit more cautious about their conclusions (e.g "is likely to have lost its function, or be only partially functional").

Details of software

For each piece of software used, please provide details of the exact version number used and any parameter settings; if default parameter settings were used, please state this. At the moment, these details are given for some of the software, but not all. For example, no version number is given for BWA, PoPoolation and qvalue; autotrim and samtools are missing details of the parameter settings, etc.

*Reviewer #3 (Recommendations for the authors):*

The genetic analysis suggests important hypotheses regarding the drought resistance or resilience of European beech. However, these hypotheses are neither very precisely articulated, nor tested in this study. Some of the information necessary to better articulate the hypotheses seems to be missing, and to articulate and test them would require substantial additional work. For this reason it seems most suitable as a Tools and Resources article accompanying publication of the dataset and improved reference genome, which could be a very valuable resource as described by the authors and the reviewers.

The leaf handling protocol is unusual and it is not very precisely reported: collected fresh from field, 50 deg C in lab 30-90 min, then to salt until storage at -80. It seems like the DNA would be better preserved if leaves were harvested directly to silica gel and not heated (in which case storage at -80 deg C can be done but is generally not required).

---

## [Author Response]

Summary:The reviewers agree that the study is of broad interest and that the genetic work represents a substantial advance on existing tools in European beech, an economically and ecologically important forest species with an uncertain future under global change. The reviewers value the case-control design using trees in standing populations and taking advantage of existing environmental data. There are questions about many important details both regarding tree selection and phenotyping, and also the genetic analysis, which are missing from the manuscript and need to be addressed. The results provide a valuable basis for the development of hypotheses regarding the adaptive potential within beech populations facing global change, but many additional steps would be required to articulate and test specific hypotheses emerging from these results. Thus we would consider a revised version of this manuscript, addressing these essential revisions and considering the other points raised, for publication as a Tools and Resources article formatted as a Short Report.

We followed the reviewers’ suggestions wherever possible, which, however, lead to a longer rather than shorter text of the manuscript

To accompany the sequencing dataset that the authors have already deposited at ENA, including an update of the beech genome (also at ENA/beechgenome.net) corresponding to the revisions made in the course of this study. The revision would need to include a full description (including specification of all data, software and parameter settings used) of how the assembly was improved, as well as details of the annotation (both genes and repeats).

The updated genome assembly is part of another project and is published separately. This paper in the publication process and already available as preprint. It contains all methodological details.

A chromosome-level genome assembly of the European Beech (*Fagus sylvatica*) reveals anomalies for organelle DNA integration, repeat content and distribution of SNPs

Bagdevi Mishra, Bartosz Ulaszewski, Joanna Meger, Markus Pfenninger, Deepak K Gupta, Stefan Wötzel, Sebastian Ploch, Jaroslaw Burczyk, Marco Thines

bioRxiv 2021.03.22.436437; doi: https://doi.org/10.1101/2021.03.22.436437

Essential Revisions:This is a list of revisions considered essential, referring to more detailed information from the individual reviews where relevant. The individual reviews are also appended in full for your information, including public reviews and a public summary designed to accompany your preprint, and recommendations for the authors from each reviewer.1) The authors state that they selected trees which were the nearest neighbors showing distinct phenotypes (apparently unaffected canopies versus canopies with dead and curled, dried leaves) and not displaying any apparent indications of insect infestation or disease which would cause such differences. This stressed versus healthy classification used by the authors is not specific to drought. To the extent that their identified genetic targets could be annotated, most of these are also generally stress responsive and not specific to drought stress. Drought is a complex stressor and it is certainly plausible that drought could be the main cause of different stress status in this study, but the authors do not present evidence to support this other than the general knowledge that beech forests suffered under the European 2018 summer drought and other recent droughts.– More direct indicators of water stress would provide better support for their classification, if any such data exist. Please refer to comments and specific suggestions from Reviewer #1 – Recommendations for authors point 1, and Reviewer #3 – Public review, "major weaknesses" section.

Please note that we presented evidence that the all trees within the 1 x 1 km grid cell of all sampling sites experienced significantly increased drought stress in the years 2018 and 2019 compared to the years before (Figure 1C). Moreover we cite two publications showing that the droughts in the years 2018 and 2019 caused severe water stress in many forest tree species (Schuldt et al. 2020a) and even more specifically lead to drought damage in German beech trees (Paar and Dammann 2019). Therefore, there can be no doubt that all trees sampled experienced substantial and unusual drought stress.

– Information on the relative health status of the chosen trees prior to the 2018 extreme drought could also be valuable, and perhaps the different climatic conditions of their study populations could be better leveraged to support the importance of drought in explaining different apparent stress status in the pairs of trees they sampled.

The categorization of each individual tree to damaged versus healthy was done opportunistically and represents only a snapshot of the acute symptoms expressed at the time of the study; we lack data on the status of these tree individuals in preceding or following years, let alone their growth or mortality. We acknowledge that in addition to drought events there could be multiple reasons for damage of branches or entire trees, e.g. attacks by insects, pathogens or tree competition, either independently or even as a response to drought. Trees are particularly vulnerable to insect or pathogen infestation in the following growing season after drought events (see Schuldt et al. 2020), hence such infestations can be both cause or effect of stress symptoms, associated with reduced defence potential. We cannot exclude that affected trees may already have experienced stress or infections in previous years, and our understanding of legacy effects of multiple symptoms is just beginning (Pflug et al. 2018, Schuldt et al. 2020). We thus reported a list of additional symptoms for a broader description of the trees’ status and their environment (Figure 2; Suppl. Table 2); and to get a broader overview of their responses, although important traits such as root damage or changes over time could not be considered. We included an additional paragraph into the discussion addressing the above issues.

In addition, we provide now data on Leaf Area Index from remote sensing for the sampled areas (thus including the sampled pairs) and correlate these with the drought stress experienced (see also previous remark) which shows a clear negative correlation of “Leaf Area Index” and “cumulated growth season evatranporation potential (Suppl. Figure 2). This shows the close relation between drought experienced and leaf loss of the woods surveyed over a longer period.

– To the extent that the authors cannot provide data on study trees which is more specific to drought stress, they must acknowledge that their stress classification cannot be clearly attributed to drought susceptibility of the trees, and may incorporate other (multiple) stress factors.

We now acknowledge in the discussion that we do not fully causative evidence that the observed phenotypes are caused by drought alone. While it is of course possible that all the sampling sites likewise experienced simultaneously another stress and that this unknown stressor alone or in conjunction with drought led to the observed stress phenotypes, it is highly unlikely that yet another stressor other than drought. The nature of this stressor that would have needed to increase simultaneously yet independently of the observed drought, is, however, completely unclear. The discussion was modified accordingly.

2) Beyond the concerns regarding interpretation of tree stress status, sufficient information must be provided regarding the trees studied.– The authors should provide evidence to support their classification of trees as healthy versus stressed (such as images – even example photographs, but better systematic photographs or aerial imagery of pairs; and/or supporting measurements; field rankings of e.g. crown transparency or apparent stress based on specific criteria).

Before going into detail, please note that any misclassification of trees due to unaccounted environmental differences would tend to dissimulate systematic genomic differences between the phenotypic classes – not enhance them. The reported results are therefore conservative in the sense that potential misclassifications may have prevented the discovery of additional associated loci, but cannot have produced spurious results. We now provide a sample of photographs trees from both phenotypic classifications, Suppl. Figure 4, which is also depicted in Figure 2 E-F.

– The authors should provide at least a rough measurement of how far pairs of trees were allowed to be apart (e.g. minimum and maximum distance in m) and roughly how many other trees were allowed to stand in between (to what extent can these trees be considered neighbors). Please refer to Reviewer #3 – Public review, "major weaknesses" section.

We provide now the mean distance (5.1 m) in the text and a distance distribution between all pairs in the Supplement. From this distance distribution, it becomes obvious that the trees were literally standing next to each other and that the root systems of most pairs overlapped and thus at least partially experienced the same soil quality, rooting depth etc. (see following comment).

– The authors must either provide information on known or expected within-site variation of water availability and other relevant factors such as soil quality, possible rooting depth, and differences in biotic interactions which could have affected drought status or responses, on the scale of distance between their selected pairs; or at a minimum, they should discuss possible environmental variation on this scale based on literature such as that suggested by Reviewer #1.

Suggested literature was included in the discussion, but see above – any such micro-heterogeneity in environmental conditions among paired trees cannot have contributed to artificially increase systematic genomic differences between them.

– There is conflicting information regarding numbers of individuals sampled, and included in each group, which must be resolved. Please refer to comments from Reviewer #2 under Recommendations for authors – Major suggestions – Number of individuals used.

We want to stress that there was no conflicting information regarding numbers. We admit that the (correct) information about the sizes of different sets might have been confusing. We tried to clarify these issues in the revision.

3) There is also important information missing regarding sample and data analysis, and this must be provided, and concerns regarding non-standard methods addressed. Please see detailed comments from Reviewer #2 (Recommendations for authors – Major suggestions) regarding the following:– Insufficient information is provided regarding the improvement of the reference genome.

The improvement of the genome is now described in a separate article:

A chromosome-level genome assembly of the European Beech (*Fagus sylvatica*) reveals anomalies for organelle DNA integration, repeat content and distribution of SNPs

Bagdevi Mishra, Bartosz Ulaszewski, Joanna Meger, Markus Pfenninger, Deepak K Gupta, Stefan Wötzel, Sebastian Ploch, Jaroslaw Burczyk, Marco Thines

bioRxiv 2021.03.22.436437; doi: https://doi.org/10.1101/2021.03.22.436437

– The authors used unevenly sized pools for poolGWAS and equal membership of pooled samples can also not be guaranteed based on the information provided in the Materials and methods.

The pools that were actually contrasted in poolGWAS were of identical size; their common analysis with a CMH test (the multivariate version of a Fisher’s exact test) is not affected by this. See also comments to reviewers.

Regarding the individual contribution to the pools, Gaultier et al. 2013 showed that even for substantial unequal contributions of each individual to the final pool of sequence reads, the estimation of allele frequencies is at least of the same accuracy as individual‐based analyses. We provide information that we used equal amounts of DNA from each individual (which was mainly assured by the already included information that we used a similar amount of tissue from each tree).

– Insufficient information is provided regarding how GWAS was conducted.

More detailed information on how the individual GWAS was conducted is now included in the Materials and Methods section and the complete pipeline added to Suppl. Info 2.

– It is not clear which indicators of statistical significance are reported and how these are to be interpreted.

Issue resolved, see comment to the reviewer.

– The reported very low LD may be an artifact of interpreting LD from poolGWAS data, and should be assessed using the individual genomes which were sequenced at high coverage.

Please note that the cited reference (Schlötterer et al. NRG 2014) does NOT call the method or accuracy of genome-wide (short range) LD estimates from PoolSeq data into question. On the contrary: the method is listed in a table for (recommended) PoolSeq software. It simply states that analyses requiring (long range) LD information should not rely on PoolSeq data, if haplotypes are longer than the read length. We have nevertheless added an average LD estimate from the individual resequenced data. The outcome is qualitatively identical, quantitatively, the individual reseq data suggested an even steeper drop of correlation among SNPs (Figure 3a)

– It is not clear how group membership prediction was conducted and how the very high success rate is to be interpreted (GP analysis).

See below.

– The functional interpretation of SNPs is complicated by some apparent contradictions in genome locations and gene models, some of which may be resolved by more information from the improved genome assembly.

The single apparent contradiction was probably caused by the page-break in the Table, that somehow let a line vanish unnoticed. This mishap then propagated further, which we beg to excuse. The incriminated position belongs to a different gene model. The table and all consequential errors were corrected.

– Insufficient information is provided regarding software versions and parameters used, and also reasoning for choosing some non-standard software tools.

Parameters and versions were exhaustively added. What are (non)standard software tools?

– Additionally, from Reviewer # 3, Recommendations for authors: The leaf handling protocol is unusual and it is not very precisely reported: collected fresh from field, 50 deg C in lab 30-90 min, then to salt until storage at -80. It seems like the DNA would be better preserved if leaves were harvested directly to silica gel and not heated as is commonly done (in which case storage at -80 deg C can be done but is generally not required).

Please note that we simply and faithfully reported each step of the sampling and leaf handling procedure. As we encountered no problems in terms of quantity and quality of the DNA with our low-cost, field-proof method, we do not see the need to justify the reported proceeding. What is meant by “not very precisely reported”? Even the telegram-style summary above would allow exact replication.

4) One important hypothesis suggested by this study is that beech populations locally embody sufficient variation in drought-responsive genes to adapt to increased drought severity and frequency. However, this cannot be supported as a conclusion of the study.

This is based in our undisputed inference that “trees with the highest predictive values showed no loss of heterozygosity [which] indicated that there is still adaptive potential for drought adaptation in the species”. We therefore provided evidence that genetic variation for this quantitative trait is not depleted and strengthen this argument with a citation. We agree that this is no conclusive evidence and needs to be confirmed in future studies, but it is also more than a mere hypothesis. By the way, this claim occurred only in the one-sentence-twitter-style Impact Statement, not the scientific parts of the text. We down-toned it to: “European beech harbours substantial genetic variation at genomic loci associated to drought resistance and the loci identified in this study can help to accelerate and monitor this process.”

The authors should re-state this as a hypothesis and propose work required to test this hypothesis rigorously. As noted by Reviewer 1, the candidate genes revealed often do not overlap with those reported from other studies, including studies in beech,

We have refuted this critique in our direct response to Reviewer 1. The majority of all genes with known function were already implicated in drought response in previous studies. It remained unclear whether the remaining genes had not yet been associated with drought, or whether we were just unable to find them due to the lack of annotation. Respective sentence inserted.

and as noted by Reviewer 3, the geographic limitations of the study preclude conclusions regarding to what extent the study populations embody the potential of members of the species to adapt to drought.

We addressed this issue by adding a respective paragraph in the discussion.

Furthermore, there is insufficient phenotypic data to link the identified loci to specific response curves (e.g. change in ability to tolerate or recover from a given severity of water deficit and associated stresses such as salinity and heat). The ability to predict membership in a group (healthy versus stressed) by genotype demonstrates the consistency of associations identified within this dataset, but it does not indicate that the study populations are resilient to predicted future increases in drought frequency and severity.

Such functional assays to link genomic variation with phenotypic response curves of more or less wild forest trees over their life cycle would slightly exceed our current possibilities. Nevertheless, there is enough current literature, including a paper suggested by a reviewer that demonstrate this possibility from genomic data (e.g. Capblancq, T., Fitzpatrick, M. C., Bay, R. A., Exposito-Alonso, M., and Keller, S. R. (2020). Genomic prediction of (mal) adaptation across current and future climatic landscapes. Annual Review of Ecology, Evolution, and Systematics, 51, 245-269.)

Again, rather, these results suggest hypotheses about how specific loci may, alone or in concert, confer resilience or tolerance to a range of drought stress, which could be better articulated based on knowledge in the beech system and gene annotation; and the authors could suggest follow-on studies, increasing the value of their own study as a resource.

The main aim of this study was to test whether there is heritable genetic variation for this quantitative trait, i.e. if the species is likely to persist under climate change by natural selection and/or assisted evolutionary management. To study physiological or molecular details on whether and how specific loci may contribute to drought resilience or tolerance was not our aim. We now acknowledge that our study can be used as starting point for such research. We added a respective sentence.

Reviewer #2 (Recommendations for the authors):This is an interesting study on the genomic basis of drought resistance in an important forest tree species. However, I have concerns about some of the analyses and there are several aspects of the paper where there is a lack of detail or where further clarification is needed. I outline these points below.Number of individuals usedI am confused about the number of individuals included in some of the analyses because there are conflicts between the details provided in different parts of the manuscript. For example, Lines 76-77 of the introduction suggest that more than 400 individuals were used for the pool-GWAS ("more than 200 neighbouring pairs of trees"), but Line 120 states that "300 beech trees" were used for the pool-seq. Lines 122-123 say that there were 100 individuals in each of the South pools, but, Table S1 indicates there are 135 trees in each South pool. On Lines 127-128 it states "All 100 individuals from the North population" were individually re-sequenced, but the Materials and Methods say 102 individuals were re-sequenced (Line 271) and Table S1 indicates that it was 95 individuals. On Lines 238-239 it says 53 trees were sampled for "set 2", but Table S1 indicates there are 80 individuals in set 2. There are various other similar examples, which altogether create a confusing picture. The Authors need to double-check all such details and correct them as necessary.

See above.

Reference genomeMore details about the methods used for improving the reference genome are needed. For example, please provide more details of how the Hi-C reads were generated (i.e. source of DNA, extraction method, sequencing coverage) and add details of the version number and parameter settings used for the allhic software. Also, was the improved assembly reannotated? If so, how? If the original annotation was transferred to the new assembly, this should also be explained.

See above.

GWASThe guidance for popoolation (https://sourceforge.net/p/popoolation2/wiki/Manual/) states that, for the experimental design, the most important thing is "the amount of DNA per individual in a single pool should be constant". In the Materials and Methods section that deals with DNA extraction (starting on Line 264), there is no mention of quantification of DNA being done to ensure that the amount of DNA contributed to a pool by each individual is equal. Please can the Authors provide details of how they quantified DNA from each individual prior to pooling to ensure they contributed equally?

See comment above. We used the same amount of tissue from each individual by cutting an equally sized piece out of the leafs and measuring DNA content and quantified it. Details now added.

The guidance also states that, "the number of individuals per pool should be similar". However, the Southern pools are much larger than the Northern pools (more than double the size according to the information in Table S1). Could the Authors comment on why they used such unevenly sized pools and what impact this might have had on the results of the GWAS?

It is important that the pools, which are actually contrasted (i.e. with the same region) have similar sizes. This is given in our study design contrasting pools with healthy and damaged pools of 100 individuals from the South and 50 individuals each from the North. The multivariate CMH test, on which the identification of associated SNPs is based, is not affected.

With regard to the potential impact of population stratification, which could create spurious associations in the GWAS, the Authors state on Lines 201-202 "As expected from previous studies, we found no population structure among the sampling sites". However, I cannot see anywhere an analysis of potential structure between all populations sampled for the main GWAS analysis (i.e. the pool-GWAS). Could the Authors comment on this further please?

We now include FST distributions for genome-wide 1 kb windows among all pools, showing that i) the FST level among all pools is very low (standard deviation includes 0) and ii) not much different among regions compared to within regions.

Also, in relation to the control for FDR, the method used by the Authors (the R package qvalue) does not calculate "corrected" p-values. Normally for GWAS, false discovery rates are used to set an appropriate p-value significance threshold to control for the number of false discoveries, rather than to correct p-values as with multiple comparison correction. I.e. a threshold can be set under which the number of false positives is deemed to be acceptable. In Figure 4 the Authors present "corrected" -log10 p-values and indicate a p-value significance threshold. I think the values in this plot are actually the q-value estimates that are generated by the qvalue package. I suggest the Authors present the original p-values from the association analysis, but indicate a significance threshold depending on the number of false positives they are willing to allow within the set of "significant" SNPs. If this changes which SNPs are considered to be significant, then clearly any subsequent analyses and results that are presented should account for this fact.

The reviewer is right that qvalue does not calculate corrected p-values. However, we actually used p.adjust with Benjamini-Hochberg correction. The reported values are therefore correct. Please excuse this confusion.

Furthermore, for the GWAS analysis using data from the re-sequenced individuals (mentioned in Lines 303-308), please provide full details for what was actually done, including any additional filtering of SNPs done in PLINK (e.g. LD pruning, which is mentioned in the legend for Figure 3, but not in the Materials and Methods) and which statistical test for association was performed. Moreover, there don't seem to be any details for this GWAS analysis in the Results section, although it is mentioned on Lines 326-327 of the Materials and Methods that some of the significant SNPs were from this GWAS were included in the SNP array.

Details are now included.

LD analysisThe Authors estimated LD in beech on the basis of a subset of the pool-seq data. This suggests a very short size of linkage block on average (c.120bp), which led the Authors to conclude that most of the significant SNPs they detected are "the actual causal variants" as they are unlikely to be linked to other SNPs. However, it has been suggested that the estimates of LD obtained from analysis of pool-seq data are not very accurate (Schlötterer et al., 2014; doi:10.1038/nrg3803). The finding that SNPs that are further than c. 120bp apart are likely to be unlinked is surprising to me given the length of linkage blocks that have been detected in some other tree species. As the Authors have whole genome sequence data for individuals from the Northern pools, I would recommend that they also use these data to estimate LD to see if this produces a result that is congruent with that based on the pool-seq data and supports their assertion that they have likely identified causative SNPs.

See above.

GP analysisThe details of the GP analysis need further clarification. In particular, I am confused by the reported level of accuracy of GP with the set of 70 variants. The Authors say "Linear discriminant analysis (LDA) correctly predicted the observed phenotype from the genotype in 91 of 92 cases (98.9%)". However, in the Materials and Methods it is reported that 80% of individuals were used in training and only 15 individuals were used in the test set. Can the Authors please make clear what they actually did to test the accuracy of the LDA analysis in discriminating trees with the different phenotypes? If only 15 trees were included in the test set, then accuracy should be reported based on the results of these. Also, was any replication done to see if results differ with different subsamples of 15 individuals (i.e. using multiple random samples of 15 individuals from the full set)? If not, then I think this would be worth considering.

Due to some communication failure, the method was wrongly reported. We used LDA as implemented in PAST 4.05 with the entire data set and post-hoc prediction. Clarified in the manuscript.

Alternatively, have the Authors considered also using some of the pool-seq data to train a GP model? This could be used to test the accuracy of prediction for the individually sequenced trees, which would provide a larger number of trees for testing. Further, the details of the number of individuals used for the GP analysis and numbers and percentages of individuals used for training and testing need to be double-checked (lines 342-346), as there seem to be some inaccuracies here.

In addition we used a new machine learning method, eSPa (Horenko 2020) which avoids the problems arising from overfitting of a small data set. Here, we had 85% classification success. Method, results and discussion accordingly modified.

Functional significance of SNPstbg-tools was used to check whether significant SNPs from the GWAS cause non-synonymous substitutions in non-coding regions. Could the Authors explain why they chose to use this tool (which is described on its GitHub page as "not in its finished version and there are surely still some bugs to be found") rather than more established software such as SnpEff (https://pcingola.github.io/SnpEff/)?

The software is now in the publication process and the manuscript is available as preprint: tbg – a new file format for genomic data Philipp Schönnenbeck, Tilman SchellSusanne Gerber, Markus Pfenninger doi: https://doi.org/10.1101/2021.03.15.435393

Also, even if the software correctly predicts the impacts of the variants, these predictions are dependant on the accuracy of the genome annotation. As noted above, there is a lack of detail about the annotation for the improved reference genome (and also no indication of the availability of the annotation file). Could the Authors comment on whether they manually checked the gene models for those cases where a significant SNP fell within a gene (i.e. results in Table 1) to see if there was any evidence for gene model errors that could impact the results?

We manually checked the gene-models. All gene-models are correct and the intron/exon borders are supported by RNA-Seq data. Please refer to the new genome publication/preprint.

For example, two variants are reported to be within gene model 7.g2350.t1, one of which causes a premature stop codon. However, the position of these variants suggests they are >15Mb apart; assuming the positions have been correctly reported, I would be surprised if this gene model is correct, especially as the best matching gene from *A. thaliana* is only c. 3kb.

Thanks for remarking this, there a line simply vanished in manuscript preparation. The incriminated position belongs to a different gene model. Results accordingly corrected.

Also, in the case of 12.g1695.t1, four variants are reported with a c. 60bp region of the gene, including one inducing a premature stop codon in individuals with the alternate allele. This leads the Authors to suggest that the gene "lost its function" in individuals that carry the non-reference version of the allele. However, another possible cause of such a pattern would be incorrect specification of intron/exon boundaries in the gene model. Have the Authors double-checked this to confirm that the variants are actually within a protein-coding region?

See above.

Even if no evidence of gene model errors is found, it cannot be concluded from the sequence data alone that the gene will be non-functional, as even with a premature stop codon a truncated product with some functionality could be formed. So, I suggest the Authors need to be a bit more cautious about their conclusions (e.g "is likely to have lost its function, or be only partially functional").

The association of these variants with the damaged trees implies at least that the resulting variants are detrimental to fitness, even if some functionality is retained. Changed to “is possible…”.

Details of softwareFor each piece of software used, please provide details of the exact version number used and any parameter settings; if default parameter settings were used, please state this. At the moment, these details are given for some of the software, but not all. For example, no version number is given for BWA, PoPoolation and qvalue; autotrim and samtools are missing details of the parameter settings, etc.

Details added.

Reviewer #3 (Recommendations for the authors):The genetic analysis suggests important hypotheses regarding the drought resistance or resilience of European beech. However, these hypotheses are neither very precisely articulated, nor tested in this study. Some of the information necessary to better articulate the hypotheses seems to be missing, and to articulate and test them would require substantial additional work. For this reason it seems most suitable as a Tools and Resources article accompanying publication of the dataset and improved reference genome, which could be a very valuable resource as described by the authors and the reviewers.

Resulting hypotheses inserted.

The leaf handling protocol is unusual and it is not very precisely reported: collected fresh from field, 50 deg C in lab 30-90 min, then to salt until storage at -80. It seems like the DNA would be better preserved if leaves were harvested directly to silica gel and not heated (in which case storage at -80 deg C can be done but is generally not required).

See above. It worked.